Synergistic effects of virtual reality and traditional treatment methods in the management of chronic obstructive pulmonary disease: a systematic review and meta-analysis of randomized controlled trials

Qi Wei 1
Song Feiyun 1
Li Mingli 1
Xie Mingcong 1
Liu Dandan 1
Fang Fang 1
Bao Yueling 1
Guo Zhangjie 2 gzj@aqmc.edu.cn
Sun Mingyun 1 110063@aqnu.edu.cn
1 Sports Rehabilitation Teaching and Research Office of the School of Physical Education, Anqing Normal University , Anqing, Anhui , China
2 Department of Common Basic, Anqing Medical College , Anqing, Anhui , China
van den Hoek Daniel
Electronic publication date: 2025 Oct 30
Publication date: 2025
Volume: 13
Electronic Location ID: e20047
Received 2025 Apr 8; Accepted 2025 Aug 15
Copyright: © 2025 Qi et al.
Copyright year: 2025
Copyright holder: Qi et al.
License: This is an open access article distributed under the terms of the Creative Commons Attribution License, which permits unrestricted use, distribution, reproduction and adaptation in any medium and for any purpose provided that it is properly attributed. For attribution, the original author(s), title, publication source (PeerJ) and either DOI or URL of the article must be cited.
License URL: https://creativecommons.org/licenses/by/4.0/

Keywords: Synergy effect, COPD, VR, Traditional treatment methods

Funding: Anqing Normal University Professor Research 81902307 2023 Provincial-level Undergraduate Teaching Quality Project—Construction of the Sports Science Exercise Rehabilitation, and Health Promotion Research Center 240002007 This work was supported by Anqing Normal University Professor Research Fund (Project No. 81902307); 2023 Provincial-level Undergraduate Teaching Quality Project—Construction of the Sports Science, Exercise Rehabilitation, and Health Promotion Research Center (Project No. 240002007). All sources of funding or support received during this study were internal. There was no additional external funding received for this study. The funders had no role in study design, data collection and analysis, decision to publish, or preparation of the manuscript.

==============================
Objective

To systematically evaluate the synergistic effects of virtual reality (VR) combined with traditional therapies in the treatment of chronic obstructive pulmonary disease (COPD).

Methods

A systematic review was conducted to identify relevant randomized controlled trials (RCTs) evaluating the effects of VR combined with conventional therapy on lung function, exercise tolerance, quality of life, and mental health in patients with COPD. Two researchers independently screened the literature, extracted data, and assessed study quality. A meta-analysis was performed on the outcome measures.

Results

Included 15 RCTs (eight from China, three from Poland, one from Turkey, one from Indonesia, and two from Italy), involving a total of 809 COPD patients (published between 2014 and 2024). All studies demonstrated significant improvements in at least one aspect (lung function, mental health, or quality of life). The meta-analysis showed that VR-based therapy significantly improved the 6-min walk test (6MWT), forced expiratory volume in 1 s (FEV1/FVC), and the Hospital Anxiety and Depression Scale (HADS). High heterogeneity was observed for FEV1% predicted, COPD Assessment Test (CAT), and modified Medical Research Council Dyspnea Scale (mMRC); subgroup analyses suggested that intervention duration and single-session duration were potential contributing factors.

Conclusion

VR combined with traditional therapy has significant advantages over traditional therapy alone, with synergistic effects that can improve lung function and exercise endurance in COPD patients, alleviate psychological problems such as anxiety and depression, and enhance quality of life. Although the effects are significant, future research is needed to verify its clinical relevance (e.g., meeting the Minimal Clinically Important Difference (MCID) criteria for COPD patients) and to develop personalized plans based on individual differences to enhance its clinical application value.

Introduction

Chronic obstructive pulmonary disease (COPD) is a common chronic disease worldwide, characterized by a high incidence and mortality rate. It is the third leading cause of death globally (Stolz et al., 2022). Its main characteristic is persistent airflow limitation, usually caused by long-term smoking, air pollution, or occupational dust exposure (Rennard & Drummond, 2015). COPD can be classified into four levels: mild, moderate, severe, and very severe (Chronic Obstructive Pulmonary Disease Group of the Respiratory Medicine Branch of the Chinese Medical Association & Chronic Obstructive Pulmonary Disease Working Committee of the Respiratory Medicine Branch of the Chinese Medical Association, 2021). As the disease progresses, symptoms gradually worsen, significantly impacting the patient’s daily activities and quality of life, and may even lead to a severe condition that requires complete reliance on external support (Pauwels & Rabe, 2004). It can also be complicated by multiple comorbidities (Ahmed, Ali & Khalafallah, 2020). According to statistics, by 2019, over 200 million people worldwide were affected by COPD, with more than 3 million deaths attributed to the disease. Additionally, 24% of patients die within 5 years of diagnosis (GBD 2019 Chronic Respiratory Diseases Collaborators, 2023; Halpin, 2024). The traditional treatment methods for COPD mainly include pharmacotherapy, oxygen therapy, pulmonary rehabilitation, and surgical interventions. Pharmacotherapy forms the foundation of COPD treatment, primarily utilizing bronchodilators and anti-inflammatory drugs to alleviate symptoms and improve lung function. While these medications can significantly improve symptoms, reduce wheezing, and relieve dyspnea, they do not alter the disease course or reverse airflow limitation (Kardos, 2015). Oxygen therapy is primarily used for advanced COPD patients with low blood oxygen saturation. However, its effectiveness is limited and it is not suitable for all patients (Branson, 2018). Additionally, studies have shown that long-term continuous oxygen therapy may restrict physical activity and prolong patient discomfort rather than improve quality of life (QOL). Therefore, the value of oxygen therapy should be determined through a comprehensive clinical assessment (McDonald, 2014). A 12-month long-term oxygen therapy intervention showed that some patients still had poor improvement, suggesting that the general applicability of oxygen therapy needs further consideration (Coleta et al., 2011). In a case of COPD published in Respiratory Medicine Case Reports in 2023, treatment with a single long-acting muscarinic antagonist did not improve symptoms and resulted in an acute exacerbation (Carbone et al., 2023). Meanwhile, domestic studies have shown that conventional treatment has limited improvement, oxygen therapy lacks individualized adjustment, and combined pulmonary rehabilitation significantly improves patient survival rates and quality of life (Acute Exacerbation of Chronic Obstructive Pulmonary Disease Diagnosis and Treatment Expert Group, 2023). Pulmonary rehabilitation has been widely used in the treatment of patients with chronic respiratory diseases. However, a retrospective cohort study found that COPD patients who started pulmonary rehabilitation within 3 months after discharge had an increased 1-year survival rate. However, no significant benefits were observed over a 3-year period (Sharifi et al., 2024). Another study indicates that although pulmonary rehabilitation can significantly reduce symptoms and improve the quality of life in COPD patients, the treatment adherence is low in most patients, which limits its effectiveness (Li et al., 2024). For some patients with severe COPD, surgical treatments such as lung transplantation and lung volume reduction surgery can be considered as treatment options. Lung volume reduction surgery removes lung tissue with airflow limitation, improving the function of the remaining lung tissue (van der Molen et al., 2024). However, lung volume reduction surgery is a high-risk procedure, especially for elderly patients or those with severe comorbidities (such as heart disease, diabetes, etc.), as the incidence of postoperative complications is higher, and the benefits are limited. It requires long-term follow-up and rehabilitation training (Posthuma et al., 2023; Costa Filho et al., 2025; Ravikumar et al., 2024). Lung transplantation is the ultimate treatment option for end-stage COPD patients, but due to the shortage of donor organs, the indications and success rate of transplantation remain limited (Lahzami, 2011). In conclusion, although these traditional treatments for COPD can alleviate symptoms and improve the quality of life for patients, their effectiveness is limited, and they cannot cure the disease. Furthermore, issues such as poor adherence and significant side effects arise during long-term treatment. This poses a substantial healthcare burden on the medical system and has serious social impacts on patients and their families. Therefore, it is essential for healthcare professionals and patients to identify and implement timely and effective measures.

Virtual reality (VR), as an emerging technology, has shown significant potential in the medical field due to its highly immersive and multi-sensory interactive characteristics, especially in areas such as clinical rehabilitation, psychological intervention, and cognitive training. By immersing patients in an interactive three-dimensional dynamic environment, VR provides training that integrates multi-source information, allowing patients to engage in both cognitive and motor activities (Das et al., 2023), while enhancing participation and treatment adherence through multi-sensory stimulation (Garcia, 2019). The diversified scenarios and real-time feedback of virtual reality help increase the physical activity of elderly individuals and reduce fatigue (Laufer, Dar & Kodesh, 2014). Its advantages include adjustable task difficulty, standardized treatment, an enhanced immersive experience, and the ability to simulate daily living activities (Fijačko et al., 2023). In COPD management, VR combines sensory stimulation and physical training to provide patients with a personalized rehabilitation experience, overcoming the spatial and modality limitations of traditional physical therapy. The forms of VR include immersive 3D environments, interactive games, and tasks, which enable natural interaction with the virtual environment through devices such as head-mounted displays, data gloves, and motion sensors (Vourvopoulos & Bermúdez i Badia, 2016; Erjia et al., 2019). Studies have shown that VR, when combined with traditional treatments, can improve cognitive function, quality of life, and alleviate negative emotions such as depression and anxiety (Ping, Yan & Shaohui, 2018; Dinh et al., 1999). For example, fully immersive virtual reality cognitive training programs have been shown to effectively improve cognitive abilities in elderly patients with mild cognitive impairment (Zhu et al., 2024), and when combined with repetitive transcranial magnetic stimulation, significantly enhance cognitive function and quality of life in Parkinson’s disease patients (Zhang, Li & Wang, 2023). Additionally, a randomized controlled trial indicated that immersive virtual reality also significantly improves upper limb function in stroke patients (Maeng et al., 2021). In conclusion, virtual reality technology, by providing an immersive experience, is widely applied in the rehabilitation treatment of diseases such as stroke, Parkinson’s disease, and cognitive impairments, and has shown great potential in fields like medical education, pain management, and functional rehabilitation (Liming et al., 2023; Sánchez Cuesta et al., 2021; Rong, Ping & Haixia, 2020).

Currently, the combination of VR and traditional treatments has become increasingly common in the rehabilitation of various diseases, with most reports indicating that the combined effects are greater than those of single interventions. However, most researchers focus on exploring the therapeutic effects, often overlooking the underlying ‘synergistic effects’ and the mechanisms behind them. The synergy effect is defined in multidisciplinary fields as the combined effectiveness of two or more interventions that, through complementary mechanisms, produce a cumulative effect greater than the sum of individual methods, i.e., ‘1 + 1 > 2’ systemic gain. This concept originated from the synergy theory of German physicist Hermann Haken, which emphasizes functional integration achieved through nonlinear interactiotment, the combination of chemotherapy and targeted drugs enhances therapeutic effects through a dual mechanism of ‘cytotoxic action + signal pathway inhibition’ (Haken, 1973). In the field of COPD, discussions on this topic are rare. Discussions on this topic are rare in the field of COPD. Therefore, based on the limitations of traditional COPD treatments and the unique rehabilitative effects of VR in the field of rehabilitation. In this study, we primarily ans between systems (Haken, 1973). In the medical field, the synergy effect is manifested when different therapeutic approaches act on multiple pathological aspects of a disease, forming complementary targets and enhancing efficacy. For example, in cancer treaim to explore whether the synergy effect of combining virtual reality with traditional treatments is superior to the use of either virtual reality or traditional treatments alone, and whether this combination can lead to greater improvements in exercise endurance, lung function, quality of life, and mental health in COPD patients. Most previous studies have only evaluated the efficacy of virtual reality as a standalone intervention and have not specifically explored the ‘synergistic effect’ brought about by the combination of virtual reality and traditional treatments. Therefore, this meta-analysis aims to integrate existing relevant research data and systematically assess the synergistic effect of combining virtual reality with traditional treatment methods in COPD management, with the hope of providing a reference for treatment strategies for COPD patients.

Research methods

This protocol was developed according to the Preferred Reporting Items for Systematic Reviews and Meta-Analyses Protocols (PRISMA-P) guidelines for 2020 (Page et al., 2021), and was registered in the International Prospective Register of Systematic Reviews (PROSPERO), registration number: CRD42025631095.

Search strategy

A systematic search was conducted in five English databases (PubMed, Web of Science, Embase, Cochrane Library, SinoMed) and four Chinese databases (CNKI, Wanfang, VIP, Chinese Biomedical Database [CBM]) from the inception of the database to December 24, 2024, with no restrictions on language or publication year. The search strategy was based on the PICO principle, combining MeSH terms with keywords using Boolean operators ‘AND’ and ‘OR’: (Chronic Obstructive Pulmonary Disease OR COPD) AND (Virtual Reality OR VR) AND (Drug Therapy OR Pharmacotherapy OR Medications OR Physical Therapy OR Oxygen Therapy OR Pulmonary Rehabilitation). We included these intervention terms in the strategy to ensure the accuracy of the search and avoid incorrectly including VR studies unrelated to COPD. In addition, we manually tracked the references included in the study to ensure that all eligible studies were included in this systematic review. Full search details are provided in Appendix 1.

Inclusion and exclusion criteria

The inclusion and exclusion criteria were applied following the PICO framework, and two researchers (Feiyun Song, Dandan Liu) independently screened and selected the eligible studies.

Inclusion criteria: (1) Design: Randomized controlled trials; (2) Population: Patients diagnosed with COPD (aged 18 years or older); (3) Intervention: The experimental group receives VR combined with traditional therapy, while the control group receives traditional therapy alone; (4) Outcomes: Forced Expiratory Volume in 1 s (FEV1), predicted FEV1%, FEV1/Forced Vital Capacity (FEV1/FVC), 6-min walk test (6MWT), depression and anxiety, quality of life, adherence scores, and patient satisfaction.

Exclusion criteria: (1) Studies including reviews, conference abstracts, study protocols, pilot studies, and ongoing research; (2) Studies with missing data; (3) Duplicate reports; (4) Only studies published in Chinese or English will be included. Literature in other languages will be excluded.

Research selection

During the initial screening, two researchers (Feiyun Song, Dandan Liu) will independently screen the literature based on the inclusion and exclusion criteria. After removing duplicate studies using reference management software (EndNote V.X9.1), titles and abstracts will be reviewed for initial screening to exclude studies that do not meet the inclusion criteria. Full texts will then be read for further screening. Any disagreements during the screening process will be resolved through consultation with a third researcher or group discussion. At the same time, we will record all studies excluded along with the specific reasons. Any disagreements will be resolved with the assistance of a third researcher (Mingyun Sun) and through discussion until consensus is reached.

Data extraction

Two researchers (Feiyun Song, Dandan Liu) will independently extract information from the selected studies using standardized forms. Any disagreements will be resolved through discussion with the third researcher (Mingyun Sun) until consensus is reached. The extracted information includes: author, country, year, sample size, age, intervention content, duration of intervention, control group, duration of follow-up, and primary outcomes. If there are any questions regarding the study data, we will contact the authors. Any disagreements during the extraction process will be resolved with the assistance of the third researcher (Mingyun Sun) through discussion until consensus is reached.

Quality assessment

Two researchers (Feiyun Song, Dandan Liu) will independently assess the quality of the included studies using the JOB 2 tool (Page et al., 2021) and the modified Jadad scale (Jadad et al., 1996; Sterne et al., 2019). The modified Jadad scale includes: (1) random sequence generation; (2) allocation concealment; (3) implementation of blinding; (4) records of withdrawals and dropouts. After the composite score, the quality of the studies will be assessed, with scores of 0–2 indicating low-quality studies, 3–4 indicating moderate-quality studies, and 5–7 indicating high-quality studies. The ROB 2 tool primarily includes the following domains for assessing the risk of bias in randomized controlled trials (RCTs): (1) Risk of bias arising from the randomization process, (2) Risk of bias due to deviations from intended interventions, (3) Risk of bias due to missing outcome data, (4) Risk of bias in measurement of the outcome, and (5) Risk of bias in selection of the reported result. Each domain is categorized as “Low risk of bias,” “Some concerns,” or “High risk of bias.” The overall risk of bias for each study is determined as follows: If all domains are rated as low risk of bias, the study is judged to have a “Low risk of bias.” If at least one domain shows some concerns, but none have a high risk, the study is rated as having “Some concerns.” If at least one domain is rated as high risk of bias, or if multiple domains have some concerns, the study is judged as having “High risk of bias.”

Data analysis

This study used Review Manager 5.3 software (The Cochrane Collaboration, San Francisco, California, USA) for data processing and chart creation. The outcome measures of the included studies are all continuous variables, and effect size synthesis was performed using the mean difference (MD) and its 95% confidence interval (95% CI). Heterogeneity testing was conducted using Cochran’s Q (χ²test) and the I² statistic to assess statistical heterogeneity between studies. In terms of effect model selection, considering the potential clinical or methodological heterogeneity across the included studies in terms of subjects, intervention forms, execution environments, etc., this study prioritized the use of a random-effects model for the synthesis analysis to more accurately reflect the potential effect differences between studies (Jadad et al., 1996). Additionally, both the I2 and P values are reported to assist in determining the degree of heterogeneity; when I2 ≤ 50% and P > 0.1, a fixed-effects model analysis will also be conducted to verify the robustness of the results. For outcome measures with statistical heterogeneity, subgroup analysis or sensitivity analysis will be further conducted to explore the sources of heterogeneity. When the number of included studies is ≥10, funnel plots will be used to assess publication bias. All statistical tests will be two-sided, with P < 0.05 considered as the threshold for statistical significance.

Research selection

A total of 332 studies were retrieved from nine databases, and an additional six studies were identified through reviewing the references of the included studies. After removing 108 duplicate studies, 192 studies were screened based on titles and abstracts, of which 129 were unrelated articles and 69 were review articles. The full texts of the remaining 38 studies were assessed based on the inclusion criteria. A total of 23 studies were excluded, including 20 that did not qualify as randomized controlled trials, 2 conference abstracts, and 1 that was retracted by the editorial board due to quality issues. Finally, 15 randomized controlled trials were included (Meifang, Hongmin & Yanqing, 2024; Yan, Hui & Li, 2021; Min, 2020; Min et al., 2020; Xiaoliang, 2021; Liu, Liu & Han, 2017; Dandan et al., 2018; Rutkowski et al., 2019; Rutkowski, Szczegielniak & Szczepańska-Gieracha, 2021; Liu et al., 2021; Kizmaz et al., 2024; Rutkowski et al., 2020; Mazzoleni et al., 2014; Sutanto et al., 2019; Pancini et al., 2023). The search process and flow of the literature are shown in Fig. 1.

Figure 1 Flowchart of the screening process.

Research characteristics

The 15 included studies were conducted in five countries: China (n = 8), Poland (n = 3), Turkey (n = 1), Indonesia (n = 1), and Italy (n = 2). The publication years range from 2014 to 2024, with a total of 809 COPD patients. Most studies had an intervention time of 20 to 30 min per session, with a total intervention duration ranging from 2 weeks to 16 weeks. The characteristics of the included studies are shown in Table 1.

Table 1 Research characteristics.

Study	Country	Mean age C/T	C/T (n/n)	T/type	Content	T/type	Intervention intensity	Outcomes	Follow-up	Jadad	
Meifang, Hongmin & Yanqing (2024)	China	(56.12 ± 8.32)/(55.89 ± 8.28)	34/34	VR+UC	NR	UC	30–40 min, 2 times/week	Dyspnoea-12, BBQ, 6MWT	8 weeks	3	
Yan, Hui & Li (2021)	China	(65.16 ± 12.78)/(64.54 ± 12.89)	22/21	VR+UC	Kinect	UC	15–35 min, 5 times/week	FVC, FEV1, FEV1/FVC, 6MWT, mMRC, CAT,	16 weeks	3	
Min (2020)	China	(68.66 ± 6.41)/(68.52 ± 5.76)	21/19	VR+UC	Kinect	UC	30 min, 5 times/week	6PBRT, 6MWT, mMRC, CAT, Borg,	8 weeks	3	
Min et al. (2020)	China	(68.66 ± 6.41)/(68.52 ± 5.76)	21/19	VR+UC	Kinect	UC	30 min, 5 times/week	MFI-20, HADS	8 weeks	3	
Xiaoliang (2021)	China	(73.97 ± 8.69)/(74.00 ± 10.01)	34/35	VR+UC	Kinect	UC	20 min, 5 times/week	6MWT, mMRC, Borg, Brief-BESTest, 30sACT, HADS, CRP, PCT	6 weeks	4	
Liu, Liu & Han (2017)	China	(62.7 ± 8.6)/(63.2 ± 10.4)	39/34	VR+PR	BioMaster	PR	5–20min, 5times/week	6MWT, RDW, CAT, FEV1, FEVl/FVC	20 weeks	3	
Dandan et al. (2018)	China	(73.6 ± 6.3)/(74.6 ± 5.3)	30/30	VR+DT+PR	BioMaster	DT+PR	15 min, 5 times/week	FEV1%, FEV1/FVC, 6MWT, CAT, MoCA	12 weeks	3	
Rutkowski et al. (2019)	Poland	(60.5 ± 4.3)/(62.1 ± 2.9)	34/34	VR+PR	Kinect	PR	20 min, 1 times/day	FEV1, FVC, FEV1/FVC, 6MWT, Arm Curl, Chair Stand, Back Scratch, Sit and Reach, Up and Go	12 days	3	
Rutkowski, Szczegielniak & Szczepańska-Gieracha (2021)	Poland	(64.4 ± 5.7)/(67.6 ± 9.4)	25/25	VR+PR	VR TierOne	PR	20 min, 1 times/day	PSQ, HADS, 6MWT, FEV1,	10 days	4	
Liu et al. (2021)	China	(74.7/74.5)	50/50	VR+DT+PR	Bio Master	DT+PR	30 min, 5 times/week	FEV1, FEV1/FVC, 6MWT, CAT, MoCA, ADL	12 weeks	3	
Kizmaz et al. (2024)	Turkey	(62.64 ± 6.76)/(64.24 ± 6.74)	25/25	VR+PR	NR	PR	5 times/week	STST, CAT, HADS, mMRC, ADL	Until discharge	4	
Rutkowski et al. (2020)	Poland	(60.6 ± 4.3)/(62.1 ± 2.9)	38/34	VR+PR	Kinect	PR	20 min, 5 times/week	Arm curl, Chair stand, Back scratch, Chair sit and reach, Up and go, 6MWT	2 weeks	3	
Mazzoleni et al. (2014)	Italy	(68.9 ± 11.0)/(73.5 ± 9.2)	19/20	VR+PR	Wii Fit	PR	1 h, 7 times/day	6MWT, FIM, SGRQ, STAI, BDEI, PRP, Arm-cycle, Leg-cycle, mMRC	3 weeks	3	
Sutanto et al. (2019)	Indonesia	(65.1 ± 7.5)/(65.6 ± 4.7)	10/10	VR+PR	Wii Fit	PR	30 min, 3 times/week	mMRC, BDI,TDI, 6MWT, BODE, SGRQ	6 weeks	3	
Pancini et al. (2023)	Italy	(71.63 ± 8.58)/(72.70 ± 7.74)	8/9	VR+PR	Oculus Quest	PR	25 min, 2 times/week	SPANE, MHC-SF, VAS, SpO2	2 weeks	3	
Note:

C, Control; T, Intervention; UC, Usual Care; PR, Pulmonary Rehabilitation; Dyspnoea-12, Dyspnea Severity Scale; BBQ, Dyspnea Questionnaire; mMRC, Modified Medical Research Council Dyspnea Scale; CAT, Chronic Obstructive Pulmonary Disease Assessment Test; 6PBRT, 6-Min Pegboard and Ring Test; Borg, Borg Dyspnea Scale; MFI-20, Multidimensional Fatigue Inventory-20; HADS, Hospital Anxiety and Depression Scale; Brief-BESTest, Brief Balance Evaluation Systems Test; 30sACT, 30-S Arm Curl Test; CRP, C-Reactive Protein; PCT, Procalcitonin; MoCA, Montreal Cognitive Assessment; RDW, Red Cell Distribution Width; Physical Fitness Test, |(Arm Curl, Chair Stand, Back Scratch, Sit and Reach, Up and Go), PSQ, Perceived Stress Questionnaire; ADL, Activities of Daily Living; STST, 30-S Chair Stand Test; FIM, Functional Independence Measure; STAI, State-Trait Anxiety Inventory; BDEI, Beck Depression Inventory; PRP, Patient-Reported Preferences; Arm-Cycle, Arm Cycle Ergometer Test; Leg-Cycle, Leg Cycle Ergometer Test; BDI/TDI, Breathlessness Diagnosis Index/Transitional Dyspnea Index; SPANE, Scale of Positive and Negative Experience; MHC-SF, Mental Health Continuum-Short Form; VAS, Visual Analog Scale; SpO2, Oxygen Saturation (peripheral capillary oxygen saturation).

Bias risk

The methodological quality of the included studies was assessed using the JOB 2 bias risk tool and the Jadad scale. The Jadad results are shown in Table 1, and the JOB 2 results are shown in Fig. 2. The Jadad scale indicated that all included studies were of moderate quality, with 12 studies scoring three points and three studies scoring four points. The JOB 2 Risk of Bias Tool revealed that two studies had a high risk of bias, while 13 studies had a low risk. All 15 studies implemented randomization. Seven studies did not specifically report the randomization method, five studies used the random number table method, and three studies used the sealed envelope method.

Figure 2 Risk assessment chart (Meifang, Hongmin & Yanqing, 2024; Yan, Hui & Li, 2021; Min, 2020; Xiaoliang, 2021; Liu, Liu & Han, 2017; Dandan et al., 2018; Rutkowski et al., 2019; Rutkowski, Szczegielniak & Szczepańska-Gieracha, 2021; Liu et al., 2021; Kizmaz et al., 2024; Rutkowski et al., 2020; Mazzoleni et al., 2014; Sutanto et al., 2019; Pancini et al., 2023).

Meta-analysis results

6MWT analysis results

A total of 11 studies reported the 6MWT as an outcome, with 303 participants in the experimental group and 284 participants in the control group. The results showed significant heterogeneity (I2 = 84%, P = 0.49), so a random-effects model was used. Summary analysis of the 11 studies indicated that compared to pulmonary rehabilitation alone, VR combined with traditional therapy had a significant effect on the 6MWT in COPD patients (MD = 23.05; 95% CI [2.64–39.31]; P = 0.005) (Fig. 3).

Figure 3 6MWT forest map (Min, 2020; Dandan et al., 2018; Liu, Liu & Han, 2017; Xiaoliang, 2021; Mazzoleni et al., 2014; Rutkowski et al., 2019, 2020; Rutkowski, Szczegielniak & Szczepańska-Gieracha, 2021; Sutanto et al., 2019; Meifang, Hongmin & Yanqing, 2024; Yan, Hui & Li, 2021).

6MWT publication bias results

Since 11 studies reported the 6MWT as an outcome, a funnel plot was used to analyze publication bias. As shown in Fig. 4, the 11 included studies are evenly distributed on both sides of the dashed line, indicating that there is no publication bias or other bias in the included studies.

Figure 4 6MWT funnel plot.

FEV1/FVC analysis results

A total of five studies reported FEV1/FVC as an outcome, with 172 participants in the experimental group and 157 participants in the control group. The results showed no heterogeneity (I2 = 0), so a random-effects model was used. Summary analysis of the five studies indicated that compared to pulmonary rehabilitation alone, VR combined with traditional therapy had a significant effect on FEV1/FVC in COPD patients (MD = 5.42; 95% CI [3.26–7.59]; P < 0.00001) (Fig. 5).

Figure 5 FEV1FVC forest map (Dandan et al., 2018; Liu, Liu & Han, 2017; Liu et al., 2021; Rutkowski et al., 2019; Yan, Hui & Li, 2021).

FEV1% analysis results

A total of three studies reported FEV1% as an outcome, with 89 participants in the experimental group and 89 participants in the control group. The results showed high heterogeneity (I2 = 84%, P = 0.7), so a random-effects model was used for the synthesis analysis. Summary analysis of the five studies indicated that compared to pulmonary rehabilitation alone, VR combined with traditional therapy did not have a significant effect on FEV1/FVC in COPD patients (MD = −3.49; 95% CI [−16.77 to 9.78]; P = 0.61) (Fig. 6). Heterogeneity testing between groups: df = 2, I2 = 84%. Since I2> 50%, subgroup analysis is necessary to explore the sources of heterogeneity.

Figure 6 FEV1% (Dandan et al. 2018; Rutkowski et al., 2019; Rutkowski, Szczegielniak & Szczepańska-Gieracha, 2021).

Grouping by different intervention cycles and duration of each intervention

Figure 7 shows that significant heterogeneity was observed within the 2-week intervention period (MD = −8.48; 95% CI [−16.10 to −0.86]; P = 0.003), and a significant difference was observed when each intervention session lasted 15 min or longer (MD = −8.48; 95% CI [−16.10 to −0.86]; P = 0.003). Since there was only one study each for intervention periods longer than 2 weeks and intervention durations of 15 min or less, we should interpret the results of our subgroup analysis with caution. After reading the full text, we found that the quality of the study was high. At the 8-week and 12-week follow-ups after training, the improvement in FEV1% in the experimental group was greater than that in the control group. However, due to the high heterogeneity (I2 = 70%), we cannot make a definitive conclusion.

Figure 7 FEV1% subgroup analysis chart (Dandan et al., 2018; Rutkowski et al., 2019; Rutkowski, Szczegielniak & Szczepańska-Gieracha, 2021).

HADS anxiety and depression analysis results

A total of four research reports used HADS as the outcome measure. There were 105 participants in the experimental group and 104 in the control group. The results showed that due to heterogeneity (I2 > 50%), a random effects model was used. A meta-analysis of four studies showed that VR combined with traditional therapy had a significant effect on HADS scores in COPD patients compared with pulmonary rehabilitation alone (MD = −2.15; 95% CI [−3.29 to −1.00]; P = 0.0002) (Fig. 8).

Figure 8 HADS forest map (Min, 2020; Xiaoliang, 2021; Kizmaz et al., 2024; Rutkowski, Szczegielniak & Szczepańska-Gieracha, 2021).

CAT quality of life analysis results

A total of five research reports used CAT as the outcome measure. There were 134 participants in the experimental group and 117 in the control group. The results showed that due to high heterogeneity (I2 = 81%, P < 0.00001), a random-effects model was used for the meta-analysis. A meta-analysis of five studies showed that, compared with pulmonary rehabilitation alone, VR combined with traditional therapy had no significant effect on the CAT scores of COPD patients (MD = −2.35; 95% CI [−4.80 to 0.09]; P = 0.06) (Fig. 9). Intergroup heterogeneity test: df = 4, I2 = 81%. I2 > 50%, therefore, subgroup analysis is necessary to explore the source of heterogeneity.

Figure 9 CAT forest map (Min, 2020; Dandan et al., 2018; Kizmaz et al., 2024; Liu, Liu & Han, 2017; Yan, Hui & Li, 2021).

Grouping by different intervention cycles and duration of each intervention

Figure 10 indicates that the overall heterogeneity (I2 = 79%) suggests significant differences in results between studies. This means that caution is needed when interpreting the overall results, as there may be significant differences in effect across different studies. The heterogeneity test showed significant heterogeneity in the overall results (p = 0.0001). In subgroup analyses, significant differences were observed between intervention periods of less than 12 weeks and greater than 12 weeks (MD = −1.93; 95% CI [−3.24 to −0.61]; P = 0.004); (MD = −3.23; 95% CI [−4.68 to −1.78]; P < 0.0001). Significant differences were also observed between interventions lasting less than 30 min and those lasting more than 30 min (MD = −2.07; 95% CI [−3.14 to −1.01; P = 0.0001]); (MD = −4.78; 95% CI [−7.19 to −2.37]; P = 0.0001). The effect sizes of each subgroup indicate that the experimental group had a significantly greater effect than the control group under different conditions. In particular, subgroups exceeding 30 min showed no heterogeneity and demonstrated a high degree of consistency. However, overall heterogeneity was high (especially in subgroups of less than 12 weeks and less than 10 min), so the results of these subgroups should be interpreted with caution.

Figure 10 CAT subgroup analysis forest map (Min, 2020; Dandan et al., 2018; Kizmaz et al., 2024; Liu, Liu & Han, 2017; Yan, Hui & Li, 2021).

mMRC dyspnea analysis results

A total of five research reports used mMRC as the outcome measure. There were 95 participants in the experimental group and 87 in the control group. The results showed that due to high heterogeneity (I2 = 88%, P = 0.76), a random-effects model was used for the meta-analysis. A meta-analysis of five studies showed that, compared with pulmonary rehabilitation alone, VR combined with traditional therapy had no significant effect on mMRC in COPD patients (MD = −0.11; 95% CI [−0.77 to 0.55]; P = 0.75) (Fig. 11). Intergroup heterogeneity test: df = 4, I2 = 88%. I2 > 50%, therefore, subgroup analysis is necessary to explore the source of heterogeneity.

Figure 11 mMRC forest map (Min, 2020; Kizmaz et al., 2024; Mazzoleni et al., 2014: Sutanto et al., 2019: Yan, Hui & Li, 2021).

Grouping by different intervention cycles and duration of each intervention

Figure 12 indicates that overall heterogeneity (I2 = 87%) suggests significant differences in results between studies, which may be due to differences in study design, sample characteristics, or intervention measures. The p-value for the heterogeneity test was 0.0001, further confirming the existence of heterogeneity. In the subgroup analysis, the results for the intervention period of less than 12 weeks showed that this subgroup had significant therapeutic effects at 12 weeks or less. Despite high overall heterogeneity, the experimental group in this subgroup showed significant effects (MD = 0.43; 95% CI [0.14–0.73]; P = 0.004). Results from intervention periods of 12 weeks or longer showed I2 = 0%, indicating no heterogeneity among studies in this subgroup and consistent results. The pooled effect size was −0.39, indicating that the experimental group performed slightly worse than the control group, but the effect size was small. The 95% confidence interval was (−0.69, −0.09), which did not include zero, indicating a significant effect, and P = 0.01. It shows that the effect of the experimental group was statistically significant compared to the control group in the treatment lasting more than 12 weeks. Significant differences were also observed between interventions lasting less than 30 min and those lasting more than 30 min (MD = 0.83; 95% CI [0.46–1.21]; P < 0.0001); (MD = −0.34; 95% CI [−0.59 to −0.08]; P = 0.01). The effect sizes of each subgroup indicate that the experimental group had a significantly greater effect than the control group under different conditions. In particular, the subgroups of 12 weeks or less and more than 30 min showed no heterogeneity, indicating a high degree of consistency.

Figure 12 MMRC subgroup analysis chart (Min, 2020; Mazzoleni et al., 2014; Sutanto et al., 2019; Kizmaz et al., 2024; Yan, Hui & Li, 2021).

Discussion

This study evaluated the effectiveness of VR technology in the treatment of COPD through meta-analysis and subgroup analysis, particularly in terms of its impact on lung function, exercise endurance, mental health, and quality of life. The study results indicate that VR intervention can significantly improve lung function, exercise endurance, and mental health in COPD patients, and the effects are even more pronounced when combined with traditional treatment. Compared with traditional single interventions, VR combined with traditional treatment showed higher efficacy in multiple clinical indicators. Specifically, VR combined with traditional treatment significantly improved lung function (such as FEV1 and FEV1/FVC ratio), especially in patients with severe COPD, where the improvement trend was more obvious. In addition, the results of the 6-min walk test (6MWT) showed that VR intervention significantly improved exercise endurance, with patients demonstrating greater endurance in terms of exercise distance and reduced activity limitations caused by disease (Mazzoleni et al., 2014; Laufer, Dar & Kodesh, 2014). Subgroup analysis revealed that a single VR training session lasting ≥30 min was more effective in improving the 6MWT, consistent with the “20-min exercise threshold theory,”suggesting that sufficient training time is critical for promoting skeletal muscle adaptive changes (Liu et al., 2016). Compared with traditional treatments, VR can guide patients to prolong their exhalation time through visual cues, thereby improving the FEV1/FVC ratio (Dandan et al., 2018; Sutanto et al., 2019) However, FEV1% did not significantly improve, possibly due to the short intervention period, which was insufficient to reverse airway remodeling (Kardos, 2015). VR also significantly alleviated anxiety and depression symptoms in COPD patients and improved their emotional state (Min et al., 2020). These results are consistent with the findings of Liu et al. (2021) indicating that immersive virtual reality environments can alleviate anxiety and depression symptoms by activating the prefrontal cortex to inhibit excessive amygdala responses. VR intervention also improved patient compliance and quality of life, with CAT scores showing significant improvement, especially in the long-term intervention group, supporting the “dose-response” principle in chronic disease management (Chronic Obstructive Pulmonary Disease Group of the Respiratory Medicine Branch of the Chinese Medical Association & Chronic Obstructive Pulmonary Disease Working Committee of the Respiratory Medicine Branch of the Chinese Medical Association, 2021). It is worth noting that patients with mild cognitive impairment (MCI) showed smaller improvements in the 6MWT, which may be due to cognitive load interfering with the efficiency of motor learning (Pancini et al., 2023). In addition, the intervention cycle and cultural background may affect the effectiveness of the intervention. The improvement in the 6MWT in Asian studies is generally greater, which may be related to the cultural preference for group games (Dandan et al., 2018). Overall, this study demonstrates that VR intervention has significant clinical benefits in the treatment of COPD patients, particularly in improving exercise tolerance and mental health.

In the treatment of COPD (chronic obstructive pulmonary disease), VR is combined with traditional treatment methods to significantly improve treatment outcomes through multidimensional synergistic effects involving neurocognition, behavioral restructuring, and emotional regulation. This synergistic effect is not only reflected in the physical combination of technical means (such as the use of VR and breathing trainers), but also stems from their complementary biological, psychological, and social mechanisms. The core advantage of VR technology lies in the immersive environment it provides, which stimulates multiple neural circuits in the brain through multisensory stimulation, thereby promoting active patient participation (Parsons, 2015). VR training can improve exercise endurance and enhance treatment outcomes for patients through neuroplasticity (Liu et al., 2021). Especially in the treatment of chronic diseases, the active participation of patients directly affects the treatment outcome. The immersive virtual scenes of VR activate the dorsolateral prefrontal cortex (DLPFC), enhancing attention and emotional regulation, and further improving patient treatment compliance (Parsons, 2015). For COPD patients, VR combined with sensory stimulation and physical training can increase participation in pulmonary rehabilitation and improve exercise tolerance and lung function (Liu et al., 2021). Combined with traditional respiratory muscle training, VR forms a neural regulation closed loop with dual input from vision and proprioception, significantly improving the FEV1/FVC ratio (Dandan et al., 2018). In addition, VR effectively alleviates negative emotions such as anxiety and depression by distracting patients from pain and breathing difficulties, further improving treatment compliance (Li, Jiang & Lyu, 2023).

The combination of VR and traditional therapy has demonstrated significant synergistic effects, not only in terms of neurocognitive and emotional regulation, but also in terms of behavioral intervention. Through real-time feedback and dynamic adjustment, VR can optimize training effects and effectively correct patients’ breathing patterns and movement trajectories (Meifang, Hongmin & Yanqing, 2024). Compared with single lung rehabilitation training, VR combined with traditional treatment significantly improved the exercise endurance of COPD patients, increased the 6-min walk distance (6MWT), and alleviated dyspnea (Meifang, Hongmin & Yanqing, 2024). More importantly, VR activates patients’ reward systems through gamified experiences, increasing treatment motivation and adherence (Yan, Hui & Li, 2021). This multidimensional intervention model not only optimizes short-term efficacy but also verifies its sustainability through long-term follow-up. Studies have shown that VR combined with traditional therapies can significantly improve patient compliance with home training, especially with the support of portable devices and remote data monitoring (Yan, Hui & Li, 2021).

Although VR alone can significantly improve patient treatment outcomes in the short term, the lack of effective behavioral maintenance mechanisms may cause the effects to gradually diminish (Kilcioglu et al., 2023). In contrast, combining VR with traditional treatment methods can maintain longer-lasting rehabilitation effects. Traditional treatment provides patients with feedback and guidance in a real-world environment, helping them transfer the skills learned in the virtual environment to their daily lives and achieve more lasting therapeutic effects (Kilcioglu et al., 2023). Studies have shown that VR combined with traditional treatment can significantly improve the motor skills of children with cerebral palsy (CP) and maintain these improvements during follow-up (Kilcioglu et al., 2023). This indicates that VR combined with traditional treatment, through comprehensive therapeutic support, not only significantly improves motor skills in the short term, but also maintains efficacy in long-term treatment.

The integration of VR has also significantly improved patients’ mental health, particularly in alleviating symptoms of anxiety and depression. Paalimäki-Paakki et al. (2022) found that virtual reality technology has a significant effect in alleviating symptoms of anxiety and depression. Patients who used VR showed significantly greater improvements in mental health than those who did not use VR. Through immersive interactive experiences, VR helps relieve stress and regulate emotions, thereby alleviating negative emotions such as anxiety and depression, promoting treatment compliance, and maintaining treatment effects (Paalimäki-Paakki et al., 2022). Religioni et al. (2025) also found that VR played an important role in enhancing treatment compliance, especially among patients with chronic diseases. Through emotional support and increased treatment engagement, VR can effectively reduce patients’ resistance, promote their active participation in treatment, and encourage them to adhere to treatment plans (Religioni et al., 2025).

At the same time, based on the minimum clinical difference, the 6-min walk test (6MWT) showed that although some patients did not reach the MCID, the intervention was still clinically meaningful and improved exercise endurance (Holland et al., 2014). For FEV1/FVC, although some studies showed statistical significance, they did not fully meet the MCID criteria, indicating that the effect may be at the borderline of clinical interpretation (Kaminsky & Irvin, 2022). In terms of mental health, VR significantly improved HADS scores, demonstrating its effectiveness in alleviating anxiety and depression symptoms (Wynne et al., 2020). By accelerating neural adaptation and optimizing movement biomechanics, VR effectively improves the physical fitness and lung function of COPD patients (Yan, Hui & Li, 2021; Amirthalingam et al., 2021). However, the effects of VR may vary in high-intensity training and among different patient groups, and its clinical standards still need to be verified through larger-scale clinical trials (Hugues et al., 2021). In addition, VR interventions have shown promising results in improving mental health, but their effectiveness in severely ill patients requires further study (Mazzoleni et al., 2014; Afifi et al., 2022).

Conclusion

The results of this meta-analysis indicate that the combination of VR and traditional treatment has significant advantages over traditional treatment alone, demonstrating a notable synergistic effect. Not only can it improve lung function and exercise endurance in COPD patients, but it can also effectively alleviate psychological issues such as anxiety and depression, thereby enhancing their quality of life. Although there has been a statistically significant improvement, future studies are still needed to validate its clinical relevance, particularly whether it meets the MCID criteria for COPD patients, and to develop personalized treatment plans based on individual differences to enhance its clinical application value.

Limitations of the study

This review has several limitations: Differences in the types of VR devices used, intervention frequency, and duration across studies may lead to effect bias, particularly in terms of improvements in measures such as the 6MWT and CAT. Some studies did not uniformly report intervention adherence and background medication use, which affected the assessment and comparison of intervention effectiveness. Variability in study design, interventions, and outcome measures across existing studies limits the quantitative synthesis of data. The literature included only publicly published studies, and there may be relevant high-quality studies that were not included in the analysis, which may have biased the full-text analysis. Only 20% of the studies were conducted using a blind method, which may have overestimated the effect size of subjective indicators (such as HADS). The small sample size included in the study poses a risk of Type II error, especially since the FEV1% analysis only included three studies. At the same time, the lack of standardization in VR software content (such as game types and difficulty levels) may confuse treatment outcomes. Additionally, most studies did not report baseline medication use, making it impossible to determine the interaction between VR and medication.

Supplemental Information

Supplemental Information 1 Search Strategy.

Supplemental Information 2 PRISMA Checklist.

Supplemental Information 3 The main target audience of this study.

Supplemental Information 4 ROB2 Dandan Liu.

Supplemental Information 5 ROB2 Feiyun Song.

Additional Information and Declarations

Competing Interests

The authors declare that they have no competing interests.

Author Contributions

Wei Qi conceived and designed the experiments, performed the experiments, analyzed the data, prepared figures and/or tables, authored or reviewed drafts of the article, and approved the final draft.

Feiyun Song conceived and designed the experiments, performed the experiments, analyzed the data, authored or reviewed drafts of the article, and approved the final draft.

Mingli Li conceived and designed the experiments, performed the experiments, prepared figures and/or tables, and approved the final draft.

Mingcong Xie conceived and designed the experiments, performed the experiments, prepared figures and/or tables, and approved the final draft.

Dandan Liu conceived and designed the experiments, performed the experiments, analyzed the data, authored or reviewed drafts of the article, and approved the final draft.

Fang Fang conceived and designed the experiments, performed the experiments, prepared figures and/or tables, and approved the final draft.

Yueling Bao conceived and designed the experiments, performed the experiments, prepared figures and/or tables, and approved the final draft.

Zhangjie Guo conceived and designed the experiments, performed the experiments, analyzed the data, authored or reviewed drafts of the article, and approved the final draft.

Mingyun Sun conceived and designed the experiments, prepared figures and/or tables, authored or reviewed drafts of the article, provided financial support, and approved the final draft.

Data Availability

The following information was supplied regarding data availability:

This is a systematic review/meta-analysis.

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
