# Peer review of "Synergistic effects of virtual reality and traditional treatment methods in the management of chronic obstructive pulmonary disease: a systematic review and meta-analysis of randomized controlled trials"

_PeerJ, doi:10.7717/peerj.20047_

## Round 0.1 · original submission · Major Revisions

**Language Note:** The review process has identified that the English language must be improved. PeerJ can provide language editing services - please contact us at [email protected] for pricing (be sure to provide your manuscript number and title). Alternatively, you should make your own arrangements to improve the language quality and provide details in your response letter. – PeerJ Staff

·

Basic reporting

This is an interesting and timely research topic. However, some points need clarification to strengthen the manuscript.
- Please provide a clinical example of traditional treatment limitations, including a real-world clinical example, which would help illustrate the practical limitations of conventional treatments and reinforce the need for alternative approaches.
- Please elaborate on the specific features of virtual treatment modalities. A clearer description of the platforms, delivery methods, and therapeutic frameworks would enhance reader understanding.

Experimental design

- Please clarify the exclusion criteria and their Relationship to study quality assessment. It is important to clarify whether the exclusion criteria were also used as part of the methodological quality assessment for study inclusion in the meta-analysis.

Validity of the findings

- Please discuss the mechanisms behind the synergistic effects. A more detailed discussion of the potential mechanisms contributing to the observed synergistic effects would be valuable. This might include aspects like improved accessibility, patient engagement, or complementary therapeutic processes.

- The limitations section should more explicitly address the heterogeneity among included studies (e.g., differences in design, intervention type, and outcome measures), as this may impact the overall conclusions. Please provide a clear explanation, The limitations section should more clearly address the variability in study design across the included papers.

Reviewer 2 ·

Basic reporting

The study aims to evaluate the synergistic effects of VR and traditional therapies in COPD. While the topic is clinically relevant, the research question is not novel, and the concept of “synergy” is not operationally defined or quantitatively explored in the analysis. The scientific justification for the work undertaken is underdeveloped. Several systematic reviews and meta-analyses on the use of virtual reality in pulmonary rehabilitation for COPD have already been published (see below). The authors do not clearly justify the need for another review or identify a specific gap in the current evidence base.

Liu, Y., Du, Q., & Jiang, Y. (2024). The effect of virtual reality technology in exercise and lung function of patients with chronic obstructive pulmonary disease: A systematic review and meta‐analysis. Worldviews on Evidence‐Based Nursing, 21(3), 307-317.

Chai, X., Wu, L., & He, Z. (2023). Effects of virtual reality-based pulmonary rehabilitation in patients with chronic obstructive pulmonary disease: A meta-analysis. Medicine, 102(52), e36702.

Patsaki, I., Avgeri, V., Rigoulia, T., Zekis, T., Koumantakis, G. A., & Grammatopoulou, E. (2023). Benefits from incorporating virtual reality in pulmonary rehabilitation of COPD patients: a systematic review and meta-analysis. Advances in Respiratory Medicine, 91(4).

The referencing is inconsistent. For example, “Error! Reference source not found.” appears multiple times, indicating incomplete or broken references, which undermines the credibility of the manuscript.

The inclusion/exclusion criteria do not specify whether studies were eligible based on language. This omission reduces transparency and raises concerns about potential language bias.

The bottom portion of Table 1, which explains study abbreviations and outcome measures, is presented in Chinese.

Please check for any grammatical, typographical, or syntax errors.

Experimental design

Despite claiming a comprehensive search across nine databases, the number of retrieved records (only 332 before deduplication) appears unusually low given the broad topic. This raises concerns about the search strategy. The third component of the search strategy: (Drug Therapy OR Pharmacotherapy OR Medications OR Physical Therapy OR Oxygen Therapy OR Pulmonary Rehabilitation) is overly restrictive and potentially unnecessary, given that the goal is to identify studies evaluating VR combined with any traditional therapy for COPD.

Quality assessment- an outdated tool was used, please use the RoB 2 tool (a revised Cochrane risk-of-bias tool for randomized trials) and update the results and figures.
Sterne, J. A., Savović, J., Page, M. J., Elbers, R. G., Blencowe, N. S., Boutron, I., ... & Higgins, J. P. (2019). RoB 2: a revised tool for assessing risk of bias in randomised trials. bmj, 366.

The basic assumption of the fixed effect model is that all the studies are expected to share a common effect (Borenstein et al., 2021). The authors used the random effect size and fixed effect size solely based on I-square. It would be necessary to include a more appropriate rationale for the choice of random or fixed effects model in this study.

There was no assessment of the certainty of the evidence. Assessing the certainty of evidence ensures that conclusions drawn from a systematic review are robust, reliable, and useful for various stakeholders in making informed decisions. Please consider the Grading of Recommendations Assessment, Development and Evaluation (GRADE) approach.

Validity of the findings

The manuscript focuses exclusively on statistical significance without discussing the clinical relevance of the findings. For instance, while a mean difference in 6MWT or FEV1/FVC may be statistically significant, it is unclear whether these changes meet established minimal clinically important differences (MCID) for COPD patients. This omission limits the interpretability and applicability of the results to clinical practice.

---

## Round 0.2 · accepted · Accept

Thank you for your efforts in addressing the reviewer comments. You have made satisfactory changes to the manuscript and it is now accepted for publication.

Reviewer 2 ·

Basic reporting

Issues addressed in this revision.

Experimental design

-

Validity of the findings

-

Additional comments

The manuscript has been substantially improved. The authors replied satisfactorily to reviewers’ comments and took into account the suggestions given.